# Tetramethylpyrazine Antagonizes the Subchronic Cadmium Exposure-Induced Oxidative Damage in Mouse Livers via the Nrf2/HO-1 Pathway

**DOI:** 10.3390/molecules29071434

**Published:** 2024-03-22

**Authors:** Xue Hu, Siqi Zhao, Ziming Guo, Yiling Zhu, Shuai Zhang, Danqin Li, Gang Shu

**Affiliations:** 1College of Veterinary Medicine, Sichuan Agricultural University, Chengdu 611130, China; 19961591868@139.com (X.H.); 18702347473@163.com (S.Z.); 18230128637@163.com (Z.G.); 18708490254@163.com (Y.Z.); shuaizhang711@163.com (S.Z.); 2College of Veterinary Medicine, Kansas State University, 1700 Denison Ave., Manhattan, KS 66502, USA

**Keywords:** cadmium (Cd), tetramethylpyrazine, liver, oxidative stress, Nrf2 pathway

## Abstract

Hepatic oxidative stress is an important mechanism of Cd-induced hepatotoxicity, and it is ameliorated by TMP. However, this underlying mechanism remains to be elucidated. To investigate the mechanism of the protective effect of TMP on liver injuries in mice induced by subchronic cadmium exposure, 60 healthy male ICR mice were randomly divided into five groups of 12 mice each, namely, control (CON), Cd (2 mg/kg of CdCl_2_), Cd + 100 mg/kg of TMP, Cd + 150 mg/kg of TMP, and Cd + 200 mg/kg of TMP, and were acclimatized and fed for 7 d. The five groups of mice were gavaged for 28 consecutive days with a maximum dose of 0.2 mL/10 g/day. Except for the control group, all groups were given fluoride (35 mg/kg) by an intraperitoneal injection on the last day of the experiment. The results of this study show that compared with the Cd group, TMP attenuated CdCl_2_-induced pathological changes in the liver and improved the ultrastructure of liver cells, and TMP significantly decreased the MDA level (*p* < 0.05) and increased the levels of T-AOC, T-SOD, and GSH (*p* < 0.05). The results of mRNA detection show that TMP significantly increased the levels of Nrf2 in the liver compared with the Cd group as well as the HO-1 and mRNA expression levels in the liver (*p* < 0.05). In conclusion, TMP could inhibit oxidative stress and attenuate Cd group-induced liver injuries by activating the Nrf2 pathway.

## 1. Introduction

Cadmium (Cd, Cadmium) is widely present in the natural environment as well as in industrial, agricultural, and other sources of pollutants [1], causing significant economic losses and also seriously jeopardizing human health [2]. Environmental cadmium pollution problems have been prevalent in most parts of the world. As early as 1984, the United Nations Environment Programme proposed 12 hazardous chemicals of global significance, with cadmium ranked first, and it was identified by the World Health Organization as a priority for the study of food contaminants [3]. Now, the U.S. Agency for Toxic Substances and Disease Registry (ATSDR) has listed it as the 6th toxic substance that endangers the health of humans and animals [4]. With the rapid development of modern industry, human exposure to cadmium is increasing year by year. Studies have shown that 63.2% of the soil in China is contaminated with cadmium, and the amount of cadmium-contaminated grain is as high as 1.46 × 10^8^ kg [5]. Cadmium and its related compounds can enter plants through water, air, and soil in the environment and can accumulate in animals and humans through the food chain, inducing oxidative cell damage and apoptosis, resulting in tissue and organ damage [6]. Cadmium poisoning also produces bone pain, spontaneous fractures, bone defects, and other hazards [7], and is severely toxic to all tissues and organs of humans and other mammals, such as the liver, kidneys, and testes [8].

Subchronic cadmium toxicity can show significant lesions in the liver, kidney, lungs, and bones, and hepatocytes and hepatic sinusoidal endothelial cells are considered to be the main target cells for cadmium exposure in the liver [9]. Liver injuries due to cadmium poisoning are mainly characterized by cirrhosis and hepatic fibrosis [10], and also cause a variety of liver diseases including necrotizing inflammation, hyperglycemia, non-alcoholic fatty liver disease (NAFLD), and non-alcoholic steatohepatitis (NASH), which ultimately leads to hepatocellular carcinoma. They can also cause reduced levels of hepatic glycogen and antioxidant enzymes [11]. Cadmium depletes large amounts of glutathione (GSH, glutathione) and inhibits antioxidant enzyme activity, leading to a large production of reactive oxygen species (ROS, reactive oxygen species), accompanied by a significant increase in lipid peroxidation products, such as malondialdehyde (MDA, Malonaldehyde) [12]. Toxicity is related to the duration of exposure, the dose of the substance, and the route of exposure [13]. Previous studies have shown that Cd-induced oxidative stress in the liver is accompanied by the activation of the Nrf2 signaling pathway [14], which is mainly characterized by oxidative damage and dysfunction.

At present, heavy metal complexing agents are not only ineffective in treating chronic Cd poisoning but also aggravate the damage, making it urgent to find effective drugs that can treat chronic Cd poisoning [15].

Chuanxiong (Rhizoma Ligustici Chuanxiong), which grows mainly in the Sichuan province of China, is one of the most widely and longest-used herbal medicines in China [16]. Tetramethylpyrazine (TMP, tetramethylpyrazine) is an alkaloid extracted from the rhizome of Ligusticum chuanxiong [17]. As an antioxidant, it can regulate oxidative stress and the production of reactive oxygen species [18], improve microcirculation, dilate small arteries, aggregate antiplatelets, act as antioxidant, and function as a calcium antagonist and antifibrotic [19]. It has a variety of pharmacological activities, such as the treatment of ischemic strokes and Alzheimer’s disease, and neuroprotective and antitumor effects. In addition, TMP has been widely used as a vasodilator [20]. Natural Chinese herbs are characterized by low toxicity, green, non-polluting, and obvious therapeutic effects. The pharmacological effects of TMP have become a research hotspot, and certain results have been achieved. However, there is still limited information about the protective effect of chuanxiong zizine against oxidative stress induced by heavy-metal exposure. Therefore, in this project, tetramethylpyrazine (TMP), the main active ingredient of Ligusticum chuanxiong, was used as a research target, which can effectively scavenge free radicals and reduce the effects of oxidative stress on the body.

It has been shown that TMP may inhibit hepatic oxidative damage by activating the Nrf2/HO-1 pathway [21]. To further investigate whether TMP can protect mice from subchronic Cd-induced liver injuries by modulating the Nrf2 signaling pathway, we modeled subchronic Cd intoxication in mice by using a Cd solution and gavaged different doses of a TMP solution. The rats were orally administered 5 mg/kg of Cd for 4 weeks to induce hepatotoxicity, and it was shown that the activities of liver-related enzymes were significantly altered, which signaled that oxidative damage occurred in the liver [22]. Meanwhile, an animal model established by an intraperitoneal injection of a cadmium chloride dose of 2 mg/kg in C57BL/6 mice every other day with a duration of 4 weeks simulated the mechanism of cadmium subchronic toxicity more satisfactorily [23]. Therefore, taking into account the administration method and the type of experimental animals, we used a gavage of 2 mg/kg of CdCl_2_ for 28 days to establish an animal model of subchronic cadmium poisoning. The detoxification mechanism of TMP on liver injuries in subchronic cadmium-poisoned mice was thoroughly investigated from the aspects of growth performance, antioxidant function, pathological tissue structure, Cd residue, and gene expression.

## 2. Results

### 2.1. TMP Inhibits Cd-Induced Weight Loss

The mice were weighed fasting on the morning of the 1st day of the official test as the initial weight of the test and were weighed fasting on the last day as the end weight of the test (results are shown in Table 1).

The results show that there was no significant difference in the beginning weight of the mice (*p* > 0.05), but there was a significant difference in the end weight of the mice (*p* < 0.05).

Compared with the CON group, the body weights of mice in the Cd and Cd + TMP groups were significantly lower (*p* < 0.05). Compared with the Cd group, the body weights of mice in the Cd + TMP group were significantly higher (*p* < 0.05). The regression relationship showed a significant quadratic increase in body weight at the end of the test in a dose-dependent relationship with increasing TMP dose (*p* = 0.010, *R*^2^ = 0.15), which was consistent with a quadratic linear regression model.

As can be seen from Table 2, the changes in organ indices of mice in the Cd + TMP group were not significantly different from any of the CON groups (*p* > 0.05) and did not conform to the linear regression model (*p* > 0.05).

### 2.2. TMP Decreases Cd Burden in the Liver

The contents of hepatic Cd are shown in Table 3. Compared with the control group, the content of hepatic Cd significantly increased after Cd exposure (*p* < 0.05). However, compared with the Cd group, the level of Cd significantly decreased (*p* < 0.05) in the liver of the Cd + TMP group. The regression relationship showed a significant quadratic decrease in dose dependence with increasing TMP dose (*p* = 0.003, *R*^2^ = 0.55).

### 2.3. TMP Attenuated Cd-Induced Hepatic and Serum Oxidative Stress

#### 2.3.1. Liver Antioxidant Biomarkers

To assess the effect of TMP on hepatic and serum oxidative stress in Cd-exposed mice, the levels or activities of GSH, T-SOD, T-AOC, and MDA were determined. As shown in Table 4, compared with the CON group, the liver T-SOD activity was significantly higher in both Cd + TMP groups (*p* < 0.05), the T-AOC level was significantly higher in the Cd + 200 mg/kg TMP group (*p* < 0.05), the liver GSH level was significantly higher in the Cd + 100 mg/kg TMP group and the Cd + 150 mg/kg TMP group, and the MDA level was significantly decreased (*p* < 0.05). The treatment of Cd alone significantly reduced the levels of GSH and T-AOC as well as the activity of T-SOD in the liver, which was accompanied by an increase in the level of MDA (*p* < 0.05). The combined application of Cd and TMP improved the changes in the above stress indices, with all liver MDAs being significantly reduced (*p* < 0.05), and the GSH, T-AOC, content and T-SOD activity significantly increased (*p* < 0.05) in the Cd + TMP group compared with the Cd group.

Regression relationships showed a significant quadratic increase in hepatic GSH (*p* = 0.002, *R*^2^ = 0.48); T-SOD (*p* = 1.818 × 10^−7^, *R*^2^ = 0.82); and T-AOC (*p* = 1.389 × 10^−3^, *R*^2^ = 0.55) in a dose-dependent relationship as the dose of TMP was increased, and the Cd + 200 mg/kg TMP group had the highest content, while the Cd + 100 mg/kg TMP group had a lower content. Hepatic MDA levels did not show a quadratic dose-dependent relationship. As the dose of TMP increased, hepatic T-SOD (*p* = 1.688 × 10^−8^, *R*^2^ = 0.83) and T-AOC (*p* = 2.189 × 10^−4^, *R*^2^ = 0.58) showed a significant linear increase in dose relationship, both of which were the highest in the Cd + 200 mg/kg TMP group.

#### 2.3.2. Serum Antioxidant Biomarkers

As can be seen from Table 5, comparing the CON group with the Cd group, subchronic Cd poisoning significantly reduced the serum GSH, T-SOD, and T-AOC contents and significantly elevated the MDA level in mice in the Cd group (*p* < 0.05). Comparing the Cd group with the Cd + TMP group, it was observed that TMP significantly elevated the serum GSH, T-SOD, and T-AOC contents and significantly decreased the MDA level in mice in the Cd + TMP group (*p* < 0.05). When the dose of TMP was 200 mg/kg, the serum T-SOD, and GSH levels of the Cd-intoxicated mice reached the highest values, which were significantly higher than those of the other groups (*p* < 0.05), and the T-AOC levels were significantly higher than those of the Cd group (*p* < 0.05). When the dose of TMP was 200 mg/kg, the serum MDA level of the Cd-intoxicated mice reached the lowest value, which was significantly lower than that of the other groups (*p* < 0.05).

Regression relationships showed that the serum T-SOD content and the GSH content of mice in the Cd + TMP group showed significant quadratic curvilinear decreases with an increasing TMP dose (*p* = 1.934 × 10^−6^, *R*^2^ = 0.59; *p* = 2.806 × 10^−7^, *R*^2^ = 0.85), and the serum MDA content showed a significant quadratic curvilinear increase in a dose-dependent relationship with an increasing TMP dose (*p* = 1.165 × 10^−3^, *R*^2^ = 0.45).

The results show a significant decrease in the GSH, T-SOD, and T-AOC content, and the MDA content increased significantly in the Cd group compared to the CON group. The GSH and T-AOC contents increased most significantly in the Cd + TMP group compared to the Cd group, the T-SOD content increased most significantly in the Cd + 150 mg/kg group and Cd + 200 mg/kg group, and the MDA content decreased most significantly.

### 2.4. TMP Alleviates Cd-Induced Liver Histopathological Lesions

By histopathological observation (Figure 1), the control group (Figure 1A1,A2) exhibited a normal liver architecture with normal orderly arranged hepatic cords and normal hepatocytes. In comparison with the control group, histopathological lesions were observed in the Cd group (Figure 1B1,B2), including disorganized hepatic cords with swelling hepatocytes, narrowed haptic sinusoids, and a few pyknotic nuclei with condensed chromatin in some hepatocytes, and the number of inflammatory cells was significantly increased compared to the control group (Figure 2). Compared with the Cd group, the above histopathological lesions were less pronounced, and the number of inflammatory cells was significantly lower in the Cd + TMP group.

Sections of the liver from mice treated with 100 mg/kg of TMP + Cd showed that the hepatocytes were more or less similar to the control (Figure 1C1,C2) and that the nuclei of hepatocytes were condensed and lysed, and a small amount of inflammatory cell infiltration was observed in the portal area. In the liver section of the mice treated with 150 mg/kg TMP + Cd, there was an infiltration of interstitial lymphocytes and neutrophils between hepatocytes (Figure 1D1,D2), and some of the hepatocytes were deformed, showing irregular sparse swelling and degeneration. The light micrographs of the livers from mice treated with 200 mg/kg of TMP + Cd showed that the hepatocytes normally appeared in their polygonal-shaped structure (Figure 1E1,E2). Their nuclei, as well as those of the control, were vesicular and displayed their normal-shaped structures.

### 2.5. TMP Reduces Liver Ultrastructural Damage in Subchronic Cd Poisoning

The ultrastructure of the mouse livers is shown in Figure 3.

Cells in the Cd group (Figure 3B1,B2) showed wrinkled nuclei with irregular shapes and condensed chromatin. The mitochondria were swollen, with obvious cristae ruptures visible in their interiors. Obvious vacuolar degeneration was produced in the stroma, and a large number of lipid droplets existed. The nuclei of the liver cells of the mice in the Cd and 100 mg/kg TMP groups (Figure 3C1,C2) were irregularly shaped, with condensed nuclear chromatin that was not distributed uniformly. Mitochondria that were spherical or rod-shaped were mostly swollen in the field of view, and most of the mitochondrial cristae were ruptured, with unclear internal structures. There was marked vacuolar degeneration in the cytoplasm. The nuclei of the mouse liver cells were irregularly shaped, with nuclear chromatin condensed and unevenly distributed. The mitochondria were spherical or rod-shaped, with a high number of swollen mitochondria occurring in the field of view. Most of the mitochondrial cristae were ruptured, with an unclear internal structure. There was obvious vacuolar degeneration in the cytoplasm.

Compared with the CON group, the shape of the nuclei was abnormal, and damage of the mitochondria was obvious. However, compared with that of the Cd group, the swelling of the mitochondria was slightly reduced. 

In the Cd and 150 mg/kg TMP groups (Figure 3D1,D2), the nuclei of the mouse liver cells displayed irregular shapes. A small number of mitochondria, which were spherical or elongated, underwent swelling, and some of the mitochondrial cristae were broken. There was obvious vacuolar degeneration in the cytoplasm. Compared with the CON group, the nuclei had an abnormal shape, and the mitochondria were obviously damaged. However, the mitochondria were no longer so swollen and structurally indistinct compared to the Cd group. 

In the Cd and 200 mg/kg TMP groups (Figure 3E1,E2), the nuclei of the liver cells appeared rounded, with a clear and intact nuclear membrane and a uniform distribution of chromatin. The mitochondria were in the shape of a round sphere or a short rod, and the internal structure was more clearly defined. A small amount of vacuole degeneration was observed in the cytoplasm. There was no significant difference in the ultrastructure of the cells compared with the CON group. Compared with the Cd group, the cell damage was greatly reduced, and the ultrastructure was more complete.

### 2.6. TMP Enhances mRNA Expression of Nrf2 and HO-1 Pathways

The mRNA expression levels of the Nrf2 signaling pathway and its downstream signaling molecule HO-1 in the mouse livers are shown in Table 6. Compared with the CON group, the expression levels of Nrf2 and HO-1 in the liver tissues of the Cd and Cd + TMP groups were significantly higher (*p* < 0.05). The expression levels of Nrf2 and HO-1 in the liver tissues of mice in the Cd + TMP group were significantly higher compared with those in the Cd group (*p* < 0.05). A linear regression analysis showed that the mRNA expression levels of Nrf2 and HO-1 showed a dose-dependent relationship of quadratic elevation with an increasing dose of TMP administration (*p* = 2.5 × 10^−11^, *R*^2^ = 0.87; *p* = 7.063 × 10^−10^, *R*^2^ = 0.87), with the highest level being in the Cd + 200 mg/kg TMP group and the lowest being in the Cd + 100 mg/kg TMP group.

## 3. Discussion

The liver is the primary target organ for Cd accumulation and toxicity, which is one of the important mechanisms of Cd-induced liver injuries [24]. Cadmium, although a metal without redox potential, causes an increase in the intracellular content of ROS and, subsequently, the oxidative damage of lipids, proteins, and DNA [25,26]. The cadmium-induced formation of superoxide anions, hydrogen peroxide, hydroxyl radical, etc., leads to a further cascade of other reactions, such as the activation of signaling pathways, induction of autophagy, apoptosis, and gene expression, indicating that oxidative stress is one of the most important mechanisms of Cd toxicity [27]. Cell membranes are attacked by free radicals, leading to instability and lipid peroxidation, leading to their disintegration [28]. TMP acts as an antioxidant that regulates oxidative stress and reactive oxygen species (ROS) production [29]. Although it has been shown so far that TMP attenuates platinum-induced oxidative stress in rat livers [30], the role of TMP in Cd-induced subchronic liver injuries in mice has not been reported. Therefore, we conducted the present study to investigate whether TMP protects against Cd-induced liver injuries and oxidative stress associated with the Nrf2 pathway in vivo.

To reduce oxidative stress and maintain the dynamic balance of free radicals in the body, the organism has evolved an antioxidant defense system, including specific antioxidant enzymes (T-SOD) and non-enzymatic antioxidants (GSH), etc. T-SOD is the primary scavenger of free radicals in organisms, which can scavenge superoxide radicals in the body and effectively attenuate lipid peroxidation on cellular biofilms [31]. GSH is a sulfhydryl-containing tripeptide that is widely present in a variety of tissues of the animal body in high levels, and its reduced level implies a decrease in the body’s antioxidant capacity and triggers oxidative stress in the body, and consequently, there is a large increase in the generation of free radicals [32]. T-AOC represents the total antioxidant capacity of the human body. MDA is a product of lipid peroxidation and is commonly used to measure the status of endogenous oxidative damage. Thus, T-SOD, GSH, T-AOC, and MDA are important parameters in the assessment of Cd-induced oxidative damage in the liver [33]. The results of the present study show that the administration of the Cd-alone group increased the accumulation of hepatic Cd; significantly increased the MDA content in the liver and serum; and significantly decreased the T-SOD, GSH, and T-AOC content, suggesting that Cd can cause hepatic injuries and oxidative stress, leading to hepatic dysfunction, which is in agreement with the results of previous studies [24,34,35]. After treatment with TMP, the content of Cd is significantly reduced in the liver, and the antioxidant defense system of the mouse organism is enhanced. The results of this experiment confirm that TMP could reduce Cd-induced oxidative stress and its associated liver injuries.

In normal liver tissues, hepatic lobules and hepatic cords are clear; hepatocytes are structurally intact, uniform in size, and polygonal, with clear nuclei; are rich in cytoplasms; and are arranged in radial strips around the central vein to form hepatic cords. Furthermore, the hepatic sinusoids contain a large number of macrophages and NK cells. Studies have shown that after long-term Cd contamination in mice, hepatocyte edema, disorganized hepatic cords, the infiltration of neutrophils and lymphocytes in hepatic lobules and confluent areas, and alterations in liver morphology and function are induced [36]. In this experiment, normal liver tissues were seen in the control group, while in the Cd group, the hepatic cord arrangement was slightly disordered, and there were neutrophils and lymphocytes infiltrating in the hepatic lobules and the confluent area, which is basically consistent with previous studies.

It has been proven that TMP, as a strong antioxidant, can scavenge reactive oxygen species, effectively improving lipid peroxidation in the organism, and can have a protective effect on a variety of liver injuries, which can improve the pathological state of liver fibrosis and inhibit the inflammatory response of liver fibrosis through a variety of pathways [37]. The results of this experiment show that compared with the Cd group, liver injuries were significantly improved in the test group, especially in the 200 mg/kg TMP + Cd group, which is consistent with the results of existing studies. This suggests that TMP treatment can improve the pathological state of hepatic fibrosis and inhibit the inflammatory response of hepatic fibrosis through multiple pathways. TMP treatment, especially high-dose TMP treatment, exhibits a significant protective effect on liver injuries in mice caused by subchronic Cd exposure.

The transmission electron microscopy results show abnormalities in the ultrastructure of the livers of mice in the Cd group, such as mitochondrial swelling and irregularities in the morphology of the nucleus, indicating the direct toxicity of CdCl_2_. Colegio et al. [38] and Waisberg et al. [39] attributed chromatin condensation to the gradual inactivation of nuclear components, which was probably due to the inhibition of DNA repair and DNA methylation. Morphological changes in mitochondria reflect an impairment of mitochondrial integrity, particularly changes in the mitochondrial membrane potential, an increase in intracellular Ca^2+^, apoptosis, cellular respiration, an inhibition of ATP synthesis, and an inhibition of oxidative phosphorylation [40].

Compared with the CON group, the mitochondria, nuclei, and cytoplasmic matrix of the mouse liver cells in the Cd group showed significant damage and degeneration, demonstrating that subchronic Cd intoxication can disrupt the energy metabolism and physiological functions of mouse hepatocytes. Studies have shown that mitochondria are one of the early targets of Cd-induced cytotoxicity, which is mainly manifested as a Cd-induced reduction in the mitochondrial membrane potential [41]. The reduction in the mitochondrial membrane potential is not only an irreversible event in the early stage of apoptosis but is also an inevitable event for the occurrence of apoptosis [42].

WEN [43] found that expression levels of the mitochondrial marker proteins HSP60 and COXIV were significantly reduced after the Cd treatment of rat cerebral cortical neurons for 12 h and 24 h. The mitochondrial ultrastructure showed obvious damage, forming a large number of mitochondrial auto-phagosomes, which suggests that Cd removes neuronal mitochondria by activating mitochondrial autophagy. This further demonstrates that Cd could promote mitochondrial autophagy and damage the mitochondrial structure by lowering the mitochondrial membrane potential, thus promoting apoptosis and causing the liver to lose its normal physiological function.

Compared with the Cd group, the number of mitochondria undergoing swelling was significantly reduced in the Cd + 100 mg/kg TMP-treatment group and the Cd + 150 mg/kg TMP-treatment group. The destruction of the internal structure of the mitochondria was alleviated, and the lipid droplets in the cells were reduced. The ultrastructure of the mouse hepatocytes was significantly improved. The treatment of 200 mg/kg of TMP significantly improved the abnormality of the hepatocytes caused by CdCl_2_. The liver remained almost normal in ultrastructure, indicating that the TMP treatment had a significant protective effect on the damage of the mouse livers caused by subchronic Cd exposure.

Nrf2, as a key transcription factor, plays an important role in oxidative stress by regulating the antioxidant defense system through mediating the expression of antioxidant enzymes and phase II detoxification enzyme genes in a cytoplasmic protein chaperone molecule-nuclear transcription factor-associated factor 2-antioxidant response element (Keap1-Nrf2-ARE) signaling pathway. HO-1 has been considered as an important antioxidant defense enzyme that catalyzes the degradation of hemoglobin to produce CO, bilirubin, and iron, thus exerting an antioxidant effect [44].

The results of the present study show that the mRNA expression levels of Nrf2 and HO-1 in the liver tissues of the Cd group were significantly higher than those of the CON group, suggesting that Cd can activate the Nrf2 signaling pathway at the gene level and can damage the antioxidant defense system. The results of this experimental study are similar to some literature reports: He [45] found that Nrf2 knockout mouse embryonic fibroblasts exhibited higher levels of ROS and Nrf2, but there were no induced expressions of NQO1 and HO-1 under basal conditions compared to a normal group of mice by the Cd contamination of normal and Nrf2 knockout mouse embryonic fibroblasts. This suggests that Cd activates the Nrf2 signaling pathway and induces the expression of factors related to the Nrf2 signaling pathway. Gong [46] treated albino rats with CdCl_2_ (20 mg/L) in free drinking water for 8 weeks and found that the mRNA and protein expression levels of Nrf2 and its downstream antioxidant molecules were significantly up-regulated in the liver tissues, activating the Nrf2 signaling pathway. However, it has also been reported in the literature that Cd can inhibit the Nrf2 signaling pathway: Han [47] showed that long-term exposure to Cd significantly reduced the level of Nrf2 mRNA expression and affected Nrf2 translation, impairing the Nrf2-mediated defense system. Therefore, the effects of Cd on the Nrf2 signaling pathway may be related to the mechanism of its bidirectional regulatory function, as well as to factors such as the animal species, the dose and mode of administration of Cd, and the duration of intoxication.

TMP is an important antioxidant that effectively scavenges free radicals and mitigates the damage caused by oxidative stress on the body. In order to investigate whether TMP regulates the Nrf2 signaling pathway to protect subchronic Cd-induced liver injuries in mice, the mRNA expression levels of the Nrf2 signaling pathway and its downstream signaling molecule, HO-1, were examined in mouse livers in this study. The results of this study show that the mRNA expression levels of Nrf2 and HO-1 were significantly up-regulated in the liver tissues of the TMP + Cd group compared with the Cd group and were higher than those of the control group. This suggests that TMP can enhance the expression of the Nrf2 signaling pathway at the gene level and can play a protective role against hepatic oxidative stress. To date, there have been no reports on whether TMP can up-regulate Nrf2 expression, thereby exerting a protective effect against Cd-induced oxidative stress injuries in the liver, but it has been shown that TMP inhibits high-glucose-induced neutrophil extracellular capture network formation by modulating the Nrf2/HO-1 pathway [48]. The mechanism of the anti-oxidative stress action of TMP may be related to the activation of the Nrf2/HO-1 pathway, which promotes the nuclear translocation of Nrf2, enhances peroxisomal activity, and activates the expression of the receptor γ coactivator [49]. Under normal physiological conditions, the expression and viability of HO-1 is low; however, under stress conditions, a significant antioxidant effect can be demonstrated by initiating the HO-1 expression via Nrf2 [50]. Therefore, we believe that TMP can exert antioxidant effects by increasing the HO-1 expression.

Therefore, we have hypothesised the protective mechanism of TMP against oxidative damage in the liver of subchronic cadmium-exposed mice, i.e., Figure 4. Hepatocytes are the target cells for oxidative damage to the liver by Cd^2+^, and mitochondria is the main organelle involved in oxidative damage. GSH is a tripeptide containing cysteine and is the main intracellular antioxidant. GSH not only decreases free radicals, but it also reduces their accumulation in the cell, as Cd compounds are formed. Thus, the cellular GSH concentration is decreased, modifying the cell redox state and altering the antioxidant system [45,51,52]. Other antioxidant enzymes that control ROS are SOD, which is involved in the reaction through which the superoxide anion radical is converted to H_2_O_2_ by dismutation, and catalase, which converts H_2_O_2_ into H_2_O and O_2_, providing a mechanism against ROS damage. However, catalase depletion may promote the accumulation and conversion of H_2_O_2_ into the hydroxyl radical, which is more reactive, leading to oxidative damage in membrane lipids, proteins, and DNA [53]. Cd^2+^ causes the generation of reactive oxygen species (ROS) by two primary mechanisms: mitochondria and NADPH oxidase. The superoxide radical (O^2−^) is dismutated by the action of the SOD, leading to hydrogen peroxide (H_2_O_2_), which is eventually biotransformed into water (H_2_O) by the action of catalase and the GSH peroxidase, with the last one leading to the production of the oxidized GSH (GSSG), and this is regenerated to GSH by the activity of the GSH reductase. This antioxidant system fails in the presence of Cd^2+^ and produces the highly toxic molecule hydroxyl radical (–OH), which damages proteins, lipids, and DNA [46]. In other words, as Cd has no redox activity, Cd-enhanced ROS elevation occurs via the suppression of free radical scavengers (such as GSH), by the inhibition of detoxifying enzymes (such as SOD, catalase, and glutathione peroxidase) and possibly as a result of other indirect mechanisms [54]. MDA is the end product of lipid peroxidation and a major marker of oxidative damage in the organism [55]. Measurements of MDA levels can directly reflect the degree of oxidative damage to the organism, which is prone to lipid peroxidation damage to cell membranes, leading to an impairment or a loss of membrane function. With the increase in oxygen-free radicals, the MDA level increases, and the oxidative damage to the cell membrane is aggravated [56]. Lipid peroxidation increases membrane fluidity, resulting in decreased membrane function. Due to the oxidative damage caused by Cd^2+^, ultimately, the T-AOC levels in the liver are reduced [57]. The ROS-sensitive nuclear factor E2-related factor 2 (Nrf2) is also activated by the Cd-generated ROS, in an attempt to combat oxidative stress in the cell [58]. This leads to a dose-dependent increase in the expression of Nrf2-downstream target genes [59], such as HO-1 in livers [60,61]. As Nrf2 is prevalent in mammalian cells, its structure contains seven homology domains, among which the epichlorohydrin-related protein homology domain contains a base zip site, which is able to bind to antioxidant response elements and induce the transcription and expression of antioxidant enzymes, such as SOD and GSH-Px and thus reduce oxidative stress damage [62]. HO-1 expression is regulated by Nrf2, which can promote heme degradation, and the degradation product, biliverdin, is an endogenous antioxidant, which plays an important role in inhibiting oxidative stress damage in the body [21].

Previous research reported that TMP is able to maintain the integrity of mitochondria and act as a reducing agent, thereby reducing the formation of reactive oxygen species and inhibiting lipid oxidation, thereby preventing oxidative damage [63]. TMP achieves the effect of reducing oxidative damage via quenching the ROS production, restoring the activity of endogenous antioxidant enzymes [64]. TMP also protects against homocysteine-induced apoptosis in HUVECs by regulating the mitochondrial function, inhibiting apoptosis, and reducing oxidative stress [65]. The Nrf2/HO-1 signaling pathway plays an important role in the organism’s resistance to oxidative stress damage produced by various external stimuli and is the most important endogenous antioxidant signaling pathway in organisms [66]. A study found that TMP (5 and 10 mg/kg, i.v.) attenuates oxidative stress and prevents I/R-induced apoptosis by activating the antioxidant effect of HO-1 [67]. The antioxidant effect of TMP mainly results from the activation of the Nrf2/HO-1 pathway in the liver, which scavenges free radicals; inhibits NF-κB; enhances the expression of glutathione peroxidase (GSH-Px), superoxidedismutase (SOD), and other associated antioxidants; and promotes the antioxidant capacity of hepatocytes [68,69]. Our study demonstrated that TMP elevated the GSH content, SOD activity, and T-AOC levels in the liver and serum; significantly reduced the MDA content and cadmium residue in the liver; and further promoted the mRNA expression of the Nrf2/HO-1 pathway. Therefore, we hypothesize that TMP may protect against liver injuries induced by subchronic cadmium exposure by further activating the Nrf2/HO-1 pathway, enhancing the GSH content of hepatic tissues, SOD activity, and T-AOC levels; decreasing the level of MDA; and enhancing the antioxidant capacity of hepatic tissues.

## 4. Materials and Methods

### 4.1. Animals and Treatments

The use of animals was approved by the Animal Protection and Use Committee of the Sichuan Agricultural University (approval No.: dyy-2021203010). Sixty four-week-old male mice from the Institute of Cancer Research (ICR) without the carriage of pathogenic bacteria and with a body weight from 18 to 22 g were used in this study (Pegatron Biotech Co., Ltd., Chengdu, China; animal license No.: SCXK (chuang) 2020-030). The mice were housed in a standard SPF facility at room temperature (22 ± 1) °C, with a relative humidity of 50–60%, were exposed to a 12 h/12 h light–dark rhythm, and were fed a standard pellet diet and water ad libitum. The subjects were fed adaptively for one week and then randomly divided into 5 groups (*n* = 12 in each group): (1) a control group (CON); (2) a cadmium (alone) group (Cd) that received 0.002 mg/mL (2 mg/kg) of CdCl_2_ [23]; (3) a group that received 0.1 mg/mL (100 mg/kg) of TMP and 0.002 mg/mL of CdCl_2_; (4) a group that received 0.15 mg/mL (150 mg/kg) of TMP and 0.002 mg/mL of CdCl_2_; and (5) a group that received 0.2 mg/mL (200 mg/kg) of TMP and 0.002 mg/mL of CdCl_2_ [70]. Previous studies in our laboratory have demonstrated that the doses of TMP used in this test are safe for mice. The TMP was dissolved in Carboxymethyl Cellulose Sodium, and the CdCl_2_ was dissolved in distilled water, and then the mice were gavaged with TMP and CdCl_2_ by gastric tubes for 28 days. The TMP + Cd groups were given the TMP one hour prior to CdCl_2_. The control group was similarly given the same volume of Carboxymethyl Cellulose Sodium. The Cd group was given the same volume of CdCl_2_. After the last gavage of CdCl_2_ for 24 h, the mice were administered 80 mg/kg of 2% pentobarbital sodium intraperitoneally, cervical dislocation was performed while the mice were anesthetized and already unconscious, and the mice were sacrificed. Blood samples were collected from the orbital venous plexus of the mice. The blood was collected in a sterile 1.5 mL EP tube without added anticoagulants, and the serum was subsequently isolated. The serum biomarkers of oxidative stress were measured. The mice were executed by the cervical dislocation method, and their liver, heart, lungs, spleen, and kidneys were removed, weighed and recorded separately. The livers were stored and used for further histopathological and biochemical analyses, including histopathological changes, ultrastructural changes, oxidative stress biomarkers, and gene expression analyses.

### 4.2. Body Weight and Organ Index

All mice were weighed after 12 h of fasting on the first day of the official test as the initial body weight of the test and were weighed after 12 h of fasting on the 28th day of the official test as the final body weight of the test. In addition, the mental status, hair luster, activity, appetite, and excretion of the mice in each group were observed during the test period, and the death of the mice was recorded.

At the end of the test, the mice were executed by the cervical decapitation method. The heart, liver, spleen, lungs, and kidneys were removed and rinsed with saline to remove the blood stains. The organ index was calculated using the formula below [71]:Organ index (%) = [Organ weight (g)/Body weight (g)] × 100%

### 4.3. Measurement of Cd Accumulation in the Liver

A total of 0.3 g of liver tissue was taken, dried to equilibrium weight, and placed in a microwave ablation tank with 5 mL of HNO_3_ and 2 mL of H_2_O_2_. After digestion, the levels of Cd were determined by atomic absorption spectrophotometry. The reference conditions for its atomic absorption spectrophotometer are as follows: wavelength, 228.8 nm; slit, 0.2 nm−1 nm; lamp current, 2 Ma–10 mA; drying temperature, 105 °C; drying time, 20 s; charring temperature, 400 °C–700 °C; charring time, 20 s–40 s; atomization temperature, 1300 °C–2300 °C; atomization time, 3–5 s; background correction for the deuterium lamp or Zeeman effect.

### 4.4. Detection of Antioxidant Biomarkers in the Serum and Liver

The extracted liver tissues were combined with a 0.9% NaCl solution to make a 10% liver homogenate and a 20% liver homogenate. Two milliliters of the treated liver homogenate were taken, and the supernatant was centrifuged at 3000 r/min for 10–15 min. According to the biochemical kit provided by the Nanjing Jianjian Bioengineering Research Institute (Nanjing, China), the GSH, MDA, T-SOD, and T-AOC contents were determined by the biochemical method, and the data were recorded according to the number of mice in each group. The specific operation procedures were in strict accordance with the instructions of the kits.

In the mice, blood was collected from the ophthalmic venous plexus 24 h after the last gavage and then centrifuged in a high-speed freezer centrifuge after 1 h. The supernatant was aspirated as serum after centrifugation and was stored at −20 °C. The processed serum was used to determine its GSH, MDA, T-SOD, and T-AOC contents using the relevant kits provided by Nanjing Jianjian Bioengineering Research Institute (Nanjing, China), and the data were recorded according to the number of mice in each group. The specific operating procedures were in strict accordance with the instructions of the kits.

### 4.5. Histopathological Examination of the Liver

The liver samples fixed in 4% paraformaldehyde were taken; dehydrated with a gradient of 75%, 85%, and 95% ethanol, transparent with xylene; embedded in paraffin; deparaffinized; and sectioned into 4–5 μm sections. The sections were stained with eosin-hematoxylin (H&E), sealed, and photographed using a pathology slide scanner to observe the histopathological features of the liver (×100/200).

### 4.6. Ultrastructural Observations

In each group, 0.2 g of the liver was taken, immersed in glutaraldehyde, and preserved at 4 °C. Ultrathin sections were made from the preserved samples according to standard protocols, and ultrastructural changes were observed using a JEM-1400-FLASH transmission electron microscope (Jeol, Tokyo, Japan).

### 4.7. Quantitative Real-Time PCR (qRT-PCR) Analysis

#### 4.7.1. RNA Extraction and Reverse Transcription

The tissue was ground in liquid nitrogen, and 1 mL of TRIzol was added per 50–100 mg of tissue and was homogenized with a homogenizer. The sample volume did not exceed 10% of the TRIzol volume. The homogenized samples were left at room temperature (15–30 °C) for 5 min. The clarified homogenate, i.e., the supernatant, was aspirated in the next step 500~600 μL, 200 μL at a time. The supernatant was mixed with 200 μL of chloroform, shaken vigorously for 30 s, and allowed to stand at room temperature for 2 min. The centrifugation was carried out for 15 min at 4 °C 12,000× *g*. The supernatant was mixed with 200 μL of chloroform and left for 2 min at room temperature with vigorous oscillation. The previous step was repeated until the supernatant was clarified, and then the supernatant was transferred to a new tube, and the RNA was precipitated in the aqueous phase with isopropanol. Half a milliliter of isopropanol was added for every 1 mL of TRIzol used, which was shaken for 30 s and left standing at room temperature for 10 min. The tubes are centrifuged at 12,000× *g* for 15 min at 4 °C. The supernatant was poured out, and 1 mL of 75% ethanol configured with DEPC water was added, which was shaken and oscillated to mix well. This was centrifuged at 12,000× *g* at 4 °C for 15 min, and the supernatant was discarded. The above operations were repeated, the supernatant was poured out, and the EP tube was placed upside down on top of the filter paper for 3–5 min. A total of 30–50 μL of ddH_2_O was added to dissolve the RNA. The RNA purity and concentration were determined by taking 2 μL of the RNA and measuring its absorbance (A) value at wavelengths of 260 and 280 nm. If it was too concentrated, it was diluted. A 40 μL reaction system on ice was prepared: add 8 μL of the RNase-free dH_2_O and 8 μL of the 5× Prime Script Buffer2 (for Real Time) to the 20 μL reaction solution with genomic DNA that is removed sequentially; mix; and then add 2 μL of the RT Prime Mix and 2 μL of the Prime Script RT Enzyme Mix I; mix gently; and incubate at 37 °C for 15 min and 85 °C for 5 s for a reverse transcription reaction.

#### 4.7.2. Real-Time Quantitative PCR

A 10 μL reaction system was prepared, with 5 μL of TB GREEN, 0.5 μL of the upstream and downstream primers listed in Table 7 (10 μmol/L), 1 μL of cDNA, 4 μL of a DNA template, and 3 μL of ddH_2_O. The standard procedure for PCR amplification was used in a two-step method: pre-denaturation at 95 °C for 30 s; PCR reaction: denaturation at 95 °C for 5 s, extension at 60 °C for 34 s, and a total of 40 cycles. At the end of the reaction, the amplification curve and melting curve of the Real-Time PCR were confirmed, the Ct value of each sample was recorded, and the relative expression of each target gene in each group was calculated by applying the 2^−ΔΔCt^ method, and β-actin was used as the internal reference gene.

### 4.8. Statistical Analysis

The experimental data were preliminarily organized in Excel and analyzed by One-way ANOVA and regression statistical analyses with multiple comparisons (Duncan′s method). All statistical analyses were performed using the SPSS software, version 25.0. Data were expressed as a mean ± standard deviation. Significant differences were considered at *p* < 0.05.

## 5. Conclusions

In conclusion, TMP supplements could mitigate liver injuries in subchronic Cd-intoxicated mice. The results show that subchronic cadmium poisoning produces significant oxidative damage to mouse livers. However, TMP not only had no adverse effect on the growth performance of the mice but also effectively improved the antioxidant capacity of the mouse livers, enhanced the expression of the Nrf2 and HO-1 signaling pathway, and then activated the antioxidant defense system of the organism. Moreover, there was a dose-dependent relationship between the therapeutic effects of TMP, and the antioxidant capacity of the mice was the strongest at a dose of 200 mg/kg of TMP.

## Figures and Tables

**Figure 1 molecules-29-01434-f001:**
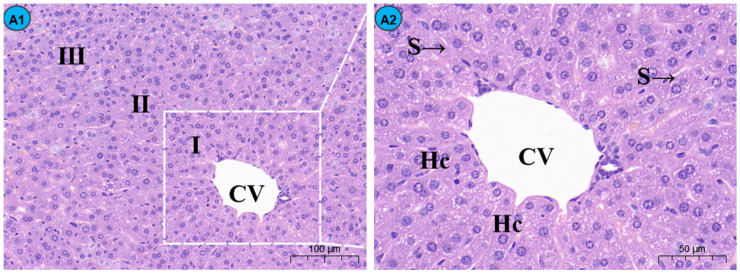
Effects of Cd and different concentrations of TMP on mouse liver histiocytes (HE staining). Notes: Representative images of the CON group (**A1**), Cd group (**B1**), Cd + 100 mg/kg TMP group (**C1**), Cd + 150 mg/kg TMP group (**D1**), and Cd + 200 mg/kg TMP group (**E1**) under a lower magnification (scale bar = 100 µm). Representative images of the CON group (**A2**), Cd group (**B2**), Cd + 100 mg/kg TMP group (**C2**), Cd + 150 mg/kg TMP group (**D2**), and Cd + 200 mg/kg TMP group (**E2**) under a higher magnification (scale bar = 50 µm). CV: central vein, S: sinusoids, Hc: hepatic cords, I: periportal zone, II: between the periportal zone and centrolobular zone, III: centrolobular zone. Black circles indicate swollen liver cells, black arrows indicate inflammatory cell infiltration, red arrows indicate nuclear consolidation and deep staining, yellow arrows indicate nuclear disintegration and disappearance, and green arrows indicate lipid droplets.

**Figure 2 molecules-29-01434-f002:**
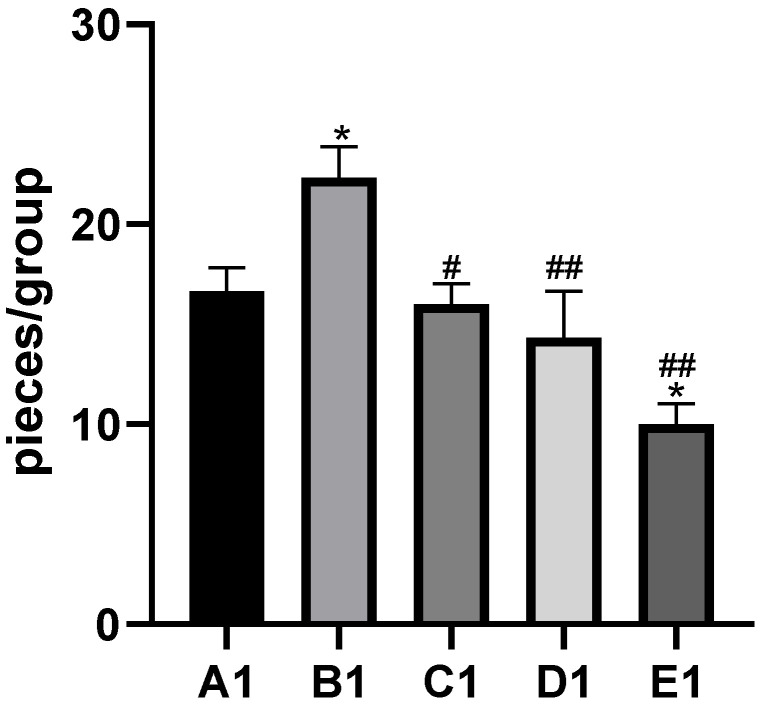
Effects of cadmium and different concentrations of TMP on the number of inflammatory cells in mouse liver histiocytes (HE staining, 10×). Notes: Representative images of the CON group (A1), Cd group (B1), Cd + 100 mg/kg TMP group (C1), Cd + 150 mg/kg TMP group (D1), and Cd + 200 mg/kg TMP group (E1) under a lower magnification (scale bar = 100 µm). * *p* < 0.05 versus group A1, # *p* < 0.05 and ## *p* < 0.01 versus group B1.

**Figure 3 molecules-29-01434-f003:**
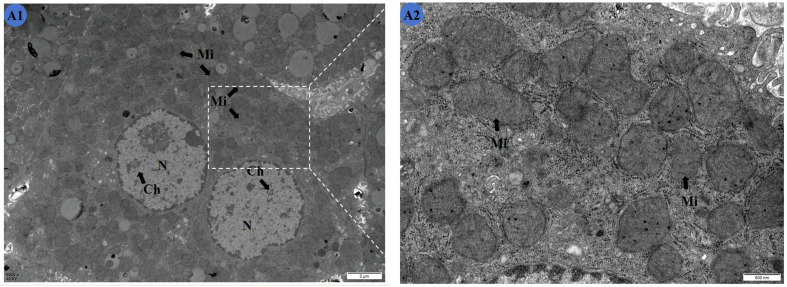
Effects of Cd and different concentrations of TMP on the ultrastructure of mouse livers. (**A1**,**A2**) Group: CON group; (**B1**,**B2**) group: Cd group; (**C1**,**C2**): Cd + 100 mg/kg TMP group; (**D1**,**D2**): Cd + 150 mg/kg TMP group; (**E1**,**E2**): Cd + 200 mg/kg TMP group. (**A1**) ×6000; (**A2**) ×24,000; (**B1**) ×6000; (**B2**) ×24,000; (**C1**) ×6000; (**C2**) ×24,000; (**D1**) ×6000; (**D2**) ×24,000; (**E1**) ×6000; (**E2**) ×24,000. N: nucleus; Ch: nuclear chromatin; ChA: nuclear chromatin condensation; Mi: mitochondria; MiS: mitochondrial swelling; L: lipid droplets. Note: Meaning of arrows: (1) black arrow: represents normal; (2) red arrow: represents abnormal.

**Figure 4 molecules-29-01434-f004:**
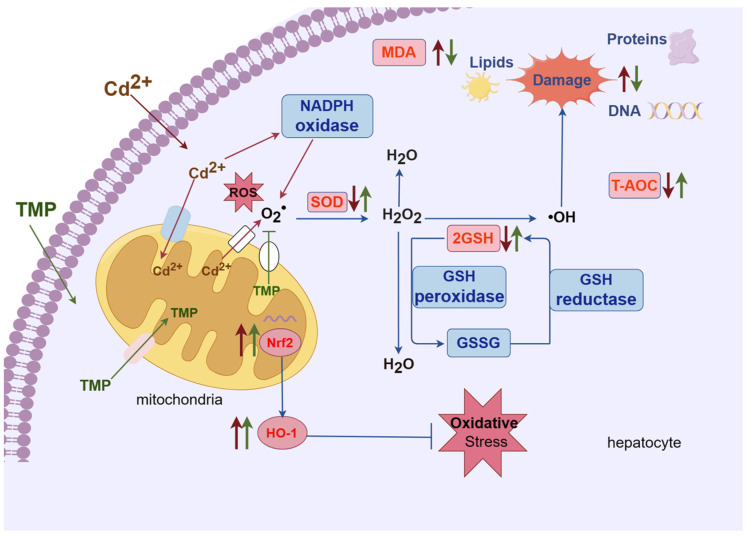
Hypothesized protective mechanism of TMP against hepatic oxidative damage in subchronic cadmium-exposed mice. Note: meaning of arrows or straight lines: (1) red thin arrow: represents Cd^2+^; (2) green thin line: TMP; (3) red thick arrow: the effect of Cd^2+^ on biochemical indices; (4) green thick arrow: the effect of TMP on biochemical indices; (5) blue thin arrow: represents the direction of the reaction and the process, no special meaning.

**Table 1 molecules-29-01434-t001:** Effects of TMP on the body weights of mice with subchronic Cd poisoning (g).

Items	CON Group	Cd Group	Cd + 100 mg/kg TMP Group	Cd + 150 mg/kg TMP Group	Cd + 200 mg/kg TMP Group	P Trend		
						Treatment Effect	Linear	Quadratic
Starting Weight (g)	27.92 ± 0.54	27.84 ± 0.50	27.92 ± 0.48	27.92 ± 0.35	27.88 ± 0.42	0.991	0.881	0.928
Final Weight (g)	32.02 ± 0.49 ^a^	29.15 ± 0.27 ^d^	29.87 ± 0.51 ^c^	30.20 ± 0.29 ^c^	30.98 ± 0.35 ^b^	<0.05	0.543	0.010

Note: Data are presented with the mean ± standard deviation (*n* = 8). abcd, different letters represent significant differences (*p* < 0.05) within the column, and the same letter or no letter represents no significant difference (*p* > 0.05).

**Table 2 molecules-29-01434-t002:** Effects of TMP on organ indices in subchronic Cd-poisoned mice (%).

Items	CON Group	Cd Group	Cd + 100 mg/kg TMP Group	Cd + 150 mg/kg TMP Group	Cd + 200 mg/kg TMP Group	P Trend		
						Treatment Effect	Linear	Quadratic
Heart Index (%)	0.60 ± 0.08	0.61 ± 0.15	0.65 ± 0.11	0.62 ± 0.18	0.99 ± 1.00	0.246	0.080	0.101
Liver Index (%)	3.98 ± 0.53	4.00 ± 0.35	3.78 ± 0.31	3.78 ± 0.31	3.85 ± 0.25	0.487	0.183	0.327
Spleen Index (%)	0.36 ± 0.10	0.34 ± 0.07	0.34 ± 0.12	0.37 ± 0.17	0.33 ± 0.12	0.930	0.760	0.930
Lung Index (%)	0.68 ± 0.10	0.64 ± 0.07	0.68 ± 0.20	0.70 ± 0.16	0.64 ± 0.09	0.702	0.887	0.885
Kidney Index (%)	1.36 ± 0.18	1.41 ± 0.18	1.41 ± 0.14	1.34 ± 0.14	1.33 ± 0.10	0.656	0.387	0.404

**Table 3 molecules-29-01434-t003:** The contents of Cd in the liver of mice exposed to Cd and/or TMP (mg/kg).

Items	CON Group	Cd Group	Cd + 100 mg/kg TMP Group	Cd + 150 mg/kg TMP Group	Cd + 200 mg/kg TMP Group	P Trend		
						Treatment Effect	Linear	Quadratic
Cd (mg/kg)	0.04 ± 0.00 ^e^	7.57 ± 0.41 ^a^	5.52 ± 0.18 ^b^	4.77 ± 0.31 ^c^	4.27 ± 0.13 ^d^	<0.05	0.241	0.003

Note: Data are presented with the mean ± standard deviation (*n* = 8). abcde, different letters represent significant differences (*p* < 0.05) within the column, and the same letter or no letter represents no significant difference (*p* > 0.05).

**Table 4 molecules-29-01434-t004:** Effects of TMP on the level of oxidative stress in the liver of Cd-intoxicated mice.

Items	CON Group	Cd Group	Cd + 100 mg/kg TMP Group	Cd + 150 mg/kg TMP Group	Cd + 200 mg/kg TMP Group	P Trend		
						Treatment Effect	Linear	Quadratic
GSH (μmol/gport)	548.90 ± 33.09 ^a^	404.47 ± 29.73 ^d^	448.08 ± 21.95 ^c^	487.06 ± 22.28 ^bc^	515.86 ± 21.61 ^ab^	<0.05	0.873	0.002
MDA (nmol/mgport)	2.92 ± 0.11 ^d^	4.34 ± 0.13 ^a^	3.25 ± 0.17 ^b^	3.17 ± 0.12 ^bc^	2.97 ± 0.10 ^cd^	<0.05	0.057	0.174
T-SOD (U/mgport)	13.51 ± 0.73 ^d^	11.40 ± 0.71 ^e^	14.70 ± 0.13 ^c^	15.63 ± 0.16 ^b^	16.51 ± 0.35 ^a^	<0.05	1.688 × 10^−8^	1.818 × 10^−7^
T-AOC (mmol/gport)	1.74 ± 0.11 ^b^	1.26 ± 0.05 ^c^	1.90 ± 0.10 ^ab^	2.05 ± 0.12 ^ab^	2.24 ± 0.42 ^a^	<0.05	2.189 × 10^−4^	1.389 × 10^−3^

Note: Data are presented with the mean ± standard deviation (*n* = 8). abcde, different letters represent significant differences (*p* < 0.05) within the column, and the same letter or no letter represents no significant difference (*p* > 0.05).

**Table 5 molecules-29-01434-t005:** Effects of TMP on serum oxidative stress levels in subchronic Cd-poisoned mice.

Items	CON Group	Cd Group	Cd + 100 mg/kg TMP Group	Cd + 150 mg/kg TMP Group	Cd + 200 mg/kg TMP Group	P Trend		
						Treatment Effect	Linear	Quadratic
T-SOD (U/mL)	220.63 ± 22.66 ^c^	181.55 ± 14.97 ^d^	223.19 ± 30.48 ^bc^	249.16 ± 22.25 ^b^	282.13 ± 20.09 ^a^	<0.05	8.558 × 10^−6^	1.934 × 10^−6^
T-AOC (U/mL)	0.84 ± 0.03 ^a^	0.73 ± 0.02 ^b^	0.77 ± 0.03 ^b^	0.84 ± 0.03 ^a^	0.86 ± 0.01 ^a^	<0.05	0.066	0.063
GSH (nmol/mL)	259.03 ± 20.02 ^d^	204.19 ± 36.55 ^e^	311.59 ± 20.73 ^c^	360.47 ± 22.10 ^b^	407.14 ± 26.17 ^a^	<0.05	2.874 × 10^−7^	2.806 × 10^−7^
MDA (nmol/mL)	16.67 ± 0.84 ^c^	20.88 ± 1.13 ^a^	18.58 ± 1.23 ^b^	17.19 ± 1.15 ^bc^	13.17 ± 1.17 ^d^	<0.05	6.167 × 10^−3^	1.165 × 10^−3^

Note: Data are presented with the mean ± standard deviation (*n* = 8). abcde, different letters represent significant differences (*p* < 0.05) within the column, and the same letter or no letter represents no significant difference (*p* > 0.05).

**Table 6 molecules-29-01434-t006:** Changes in mRNA expression levels of factors related to Nrf2/HO-1 signaling pathway in mouse livers.

Items	CON Group	Cd Group	Cd + 100 mg/kg TMP Group	Cd + 150 mg/kg TMP Group	Cd + 200 mg/kg TMP Group	P Trend		
						Treatment Effect	Linear	Quadratic
HO-1	1.09 ± 0.03 ^e^	1.19 ± 0.03 ^d^	1.26 ± 0.04 ^c^	1.36 ± 0.05 ^b^	1.54 ± 0.09 ^a^	<0.05	1.027 × 10^−11^	2.5 × 10^−11^
Nrf2	1.08 ± 0.05 ^e^	1.17 ± 0.04 ^d^	1.28 ± 0.03 ^c^	1.34 ± 0.04 ^b^	1.42 ± 0.03 ^a^	<0.05	4.629 × 10^−11^	7.063 × 10^−10^

Note: Data are presented with the mean ± standard deviation (*n* = 8). abcde, different letters represent significant differences (*p* < 0.05) within the column, and the same letter or no letter represents no significant difference (*p* > 0.05).

**Table 7 molecules-29-01434-t007:** Amplification primers for each target gene.

Gene	Accession Number	Primer Sequence (5′-3′)	Product Length (bp)
Nrf2	NM-O10902.4	F:AGCGGTAGTATCAGCCA R: GCCCAGTCCCTCAATAGC	150
HO-1	NM-O10442.2	F: TGTTGCGCTCTATCTCC R: GTACACATCCAAGCCGAG	136
β-actin	NM-007393	F: CGCTCGTTGCCAATAGTG R: GCTGTGCTATGTTGCTCTAG	117

## Data Availability

Data are contained within the article.

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
