# Peer review of "Tetramethylpyrazine Antagonizes the Subchronic Cadmium Exposure-Induced Oxidative Damage in Mouse Livers via the Nrf2/HO-1 Pathway"

_molecules, 2024, doi:10.3390/molecules29071434_

Round 1

Reviewer 1 Report

Comments and Suggestions for Authors

The authors demonstrated that sub-chronic Cd exposure damaged  mice liver, in terms of oxidative stress, histological and ultra-structures and mRNA of nrf2 and HO1, and they showed potential protection by TMP.

There are a number of concerns that the authors need to address:

1) Introduction: Global Cd pollution severity should be mentioned with a reference. General pharmacology of TMP should be mentioned more, with a few more references.

2) The major flaw throughout the Results section was the lack of a control, that is, TMP alone (at least should include TMP 200mg/kg alone). Otherwise, results are difficult to interpret, that is, TMP at such high dose may have an effect of its own, not purely a protection against Cd toxicity.

3) Related to above, about Table 5, Cd + 200 TMP group was significantly much more different from the control group; TMP on its own supposedly altered these parameters a lot.

4) The results  of Fig. 1 and 2 should be somehow quantified.

5) In section 2.6, second line, should not be Fig.10, should be Table 6. There is no Fig.10.

Comments on the Quality of English Language

need some revision

Author Response

Dear Reviewer of review,

We feel great thanks for your professional review work on our article. As you are concerned, there are several problems that need to be addressed. According to your nice suggestions, we have made extensive corrections to our previous draft, the detailed corrections are listed below:

  1. Thanks for your professional advice, we have improved the information related to the current status of global cadmium pollution and detailed the extent of the present-day hazards of cadmium pollution. At the same time, we have improved the general pharmacological effects of TMP and added more references.
  2. Thank you very much for your correction, we investigated the effects of 50, 100, 150, 200 and 250 mg/kg body weight of tetramethylpyrazine alone on the growth performance, blood physiological and biochemical indexes and antioxidant function of mice in the previous experiments, and proved that tetramethylpyrazine could significantly improve the anticoagulant and antioxidant ability of the organism, and the effect of enhancing the antioxidant ability was the most significant at the administration concentration of 150 mg/kg.
  3. It is true that the elevation of serum antioxidant function of tetramethylpyrazine in subchronic cadmium-intoxicated mice in this article alone cannot effectively exclude the possibility that the high dose of TMP has its own effect, but we have already done experiments on the elevation of antioxidant function of tetramethylpyrazine in mice, and it was proved that 150 mg/kg of tetramethylpyrazine had a good antioxidant effect on mice. However, this study demonstrated that compared with 150 mg/kg, 200 mg/kg tetramethylpyrazine had a stronger antioxidant effect on subchronic cadmium-poisoned mice, and significantly reduced liver cadmium content and the number of inflammatory cells in liver tissues, which is sufficient to prove that the high dose of tetramethylpyrazine has a protective effect against cadmium toxicity.
  4. Thanks for your suggestion, the number of inflammatory cells in each group of liver tissue has been quantified as a bar graph based on the section areas (HE staining 10×) of each group provided within the paper.
  5. Figure 10 in the second line of section 2.6 has been replaced by table 6.

we sincerely thank you for your valuable feedback that we used to improve the quality of our manuscript. Our response is given in normal font and changes to the manuscript are given in the yellow mark. If there are any other modifications we could make, we would like very much to modify them and we really appreciate your help.

Reviewer 2 Report

Comments and Suggestions for Authors

The manuscript presents results of a study demonstrating conclusively that tetramethyl pyrazine protects mice against effects of subchronic cadmium exposure, especially alleviating oxidative stress.

I do not have reservations to the design of the experiment. The description of methods requires some explanations.

There are some errors in the text that require correction.

Line 22: “the antioxidant capacity of the mice” is a too broad term, not in use; please refer to a specific tissue

Line 90: “cd-exposed”, please change to “Cd-exposed”

Tables 4 and 5: GSH is a substance, not an enzyme so its content/concentration should be expressed in mass units or moles and not in U. The same refers to AOC. Otherwise, please provide the definition of U for GSH and AOC in the Nanjing Jianjian Bioengineering Research Institute biochemical tests.  

Lines 237-239: “Metallic Cd produces oxidative damage through peroxidation, where Cd does not produce free radicals directly but superoxide, nitric oxide and hydroxyl radicals, which leads to oxidative stress-mediated injury the meaning of this sentence is unclear. Moreover, Cd was administered in the form of a salt, not as a metal.

Line 327: “catalyzes the degradation of hemoglobin to produce CO, bilirubin and iron,”, in fact heme oxygenase catalyzes the degradation of heme, not hemoglobin to produce CO, biliverdin and iron

Lines 417-420: if the blood was anticoagulated, the supernatant should be referred to as plasma, not serum

Comments on the Quality of English Language

Language not bad but some phrases require optimization

Author Response

Dear Reviewer of review,

We feel great thanks for your professional review work on our article. As you are concerned, there are several problems that need to be addressed. According to your nice suggestions, we have made extensive corrections to our previous draft, the detailed corrections are listed below:

  1. Thank you very much for your suggestion, I have elaborated the experimental methodology in more detail in the manuscript.
  2. Thank you very much for pointing out our mistake. We understand that you meant to change "antioxidant capacity of mice" to "antioxidant capacity of mouse liver" in line 22. We have sorted out our thoughts and made more changes to the abstract section, and "mouse antioxidant capacity" has been deleted.
  3. Thank you for your careful advice! We have changed "cd-exused" to "Cd-exused" in line 90.
  4. Thank you very much for your valuable suggestions, we have revised the units of GSH, T-AOC, MDA, SOD in Table 4 and Table 5.
  5. Thank you very much for your correction, we have elaborated the mechanism of oxidative damage produced by cadmium in lines 237-239 of the original article in more detail than before.
  6. Thank you very much for your correction, we have corrected "catalyzes the degradation of hemoglobin to produce CO, bilirubin and iron," in line 327 of the original article.
  7. Thank you very much for your correction, we have removed the reference to "use of anticoagulants" from lines 417-420 of the original article.

we sincerely thank you for your valuable feedback that we used to improve the quality of our manuscript. Our response is given in normal font and changes to the manuscript are given in the yellow mark. If there are any other modifications we could make, we would like very much to modify them and we really appreciate your help.

Reviewer 3 Report

Comments and Suggestions for Authors

The authors aimed at evaluating the dose-dependent (100, 150 and 200 mg.kg-1) ameliorating effect of 2,3,5,6-Tetramethylpyrazine (TMP; Pubchem CID 14296, CAS 780; XLogP3= 1.3; GRAS-flavoring agent) on cadmium (Cd)-induced (2 mg.kg-1 CdCl2; doi: 10.16303/j.cnki.1005-4545.2013.05.006) hepatotoxicity in C57BL/6J mice (n=60, ). The authors were able to demonstrate the dose-dependent reduction of TMP on subchronic hepatotoxicity (via reduction of cellular oxidative stress, histopathological, analytical, and molecular markers) but without important effects of damage in other organs. However, results and discussion are quite succinct and descriptive>inductive. Authors  are asked to make some changes in their manuscript to improve its scientific soundness and uniqueness:

·         General. A) Please be consistent with all abbreviations throughout the manuscript, including their meanings the first time they are mentioned. B) The comprehension of the manuscript could be further improved if it is proofread by a native English-language colleague or if it is sent to a formal translation agency.

·         Title. OK.  

·         Abstract. This section should be succinct without sacrificing important experimental data (including p-values).

·         Introduction. The last paragraph should include comparative information with previous studies (particularly, include more details from reference 30, justifying the used Cd dose) on this subject (e.g. doi: 10.1016/j.etp.2009.03.010), highlighting the uniqueness and scientific contribution of this new report.

·         Methods. A) Describe the units in their most common form (e.g. mg/mL = mg.mL-1). Authors should provide more details as to the “gavage” procedure used since unsupervised procedures may have had influence on results (e.g. oral mechanical damage may result in lower food consumption and so body weight reduction). B) Details of the ethical experimental approval is needed

·         Figures & Tables. A) Include a graphical abstract depicting the complete experimental design. B) Including a graphical abstract (or conclusive image) summarizing the most plausible ameliorating mechanisms of TMP on Cd-induced cytotoxicity, is strongly advised (See example in doi: 10.1080/15376516.2023.2171824). C) Figures should be provided with enough resolution (≥300 dpi) and tables should be formatted according to this journal´s guidelines. D) Instead of using primary/secondary linear use “linear” and “quadratic” and report p-values as p-trend.

·         Results & discussion. A) It is recommended to hypothesize mechanisms (evidence-based ), considering that the observed physio-morphological changes, although statistically significant due to the number of experimental units per group (n=12), are actually slight from a pathophysiological perspective. B) Include hypotheses about the molecular mechanisms associated with both Cd and TMB. A graphical explanation is strongly advised.

·         References. A) Reduce the number of referrals older than 10 years to say 20-30% (currently 48%, 15/31). B) Please check once again the references´ format according to this journal´s guidelines. C) It seems that important references on the addressed subjects were not included that could be useful for explaining the underlying mechanisms of both Cd and TMP (e.g. doi: 10.1016/j.cotox.2019.12.002, 10.3390/antiox10050630, 10.1016/j.biopha.2022.113005, 10.3390/toxics8030063, 10.1016/j.scitotenv.2023.168352, 10.1016/j.jep.2015.09.034, 10.1016/j.chemosphere.2021.129735).

Comments on the Quality of English Language

Moderate English editing is required

Author Response

  1. Dear Reviewer of review,

    We feel great thanks for your professional review work on our article. As you are concerned, there are several problems that need to be addressed. According to your nice suggestions, we have made extensive corrections to our previous draft, the detailed corrections are listed below:

    1. Thanks for your comments, we have checked the full text abbreviations and revised them.
    2. Thank you very much for your valuable suggestions, we have streamlined the abstract and presented important experimental data (including p-values).
    3. Thanks for the correction, we mentioned more previous related studies in the introduction part, and illustrated the rationality of the cadmium dose used in this study and the uniqueness of the experimental procedure by enriching the details.
    4. Thank you very much for your careful corrections, we have modified the units in the experimental methods section to show more details of the gavage procedure. All operators involved in the experiment were very experienced and careful in gavage. The operators checked the health of the mice every day during the experiment. The ethical approval number we have presented in the test methods section. We are also very conscious of the health and welfare issues of the test animals.
    5. Thanks to your suggestion, we have redesigned the full trial chart summary.
    6. Thank you for your careful corrections, we rechecked the experimental images, and the electron microscopy images in the manuscript were at sufficient resolution, although we provided liver section images at sufficient resolution (≥300 dpi), and the table formatting was modified according to journal guidelines.
    7. We have modified "Primary Linear" and "Secondary Linear" and reported p-values in terms of p-trend.
    8. Thanks to your suggestions, we have hypothesised the molecular mechanisms associated with Cd and TMB at the cellular level and have provided images in the manuscript.
    9. Thank you for your valuable suggestion, we have fulfilled the requirement by trimming the number of references that are more than 10 years old. We have also checked and corrected the formatting of the references as per the requirements of the journal. The references you have provided have been of great help to us, thank you again for your careful suggestions!

    we sincerely thank you for your valuable feedback that we used to improve the quality of our manuscript. Our response is given in normal font and changes to the manuscript are given in the yellow mark. If there are any other modifications we could make, we would like very much to modify them and we really appreciate your help.

Round 2

Reviewer 1 Report

Comments and Suggestions for Authors

Revision is satisfactory

Reviewer 3 Report

Comments and Suggestions for Authors

Thank you for adressing most (if not all) of my suggestions; the manuscript improved significantly

Comments on the Quality of English Language

Acceptable translation